# Faecal Glucocorticoid Metabolites and H/L Ratio Are Related Markers of Stress in Semi-Captive Asian Timber Elephants

**DOI:** 10.3390/ani10010094

**Published:** 2020-01-06

**Authors:** Martin W. Seltmann, Susanna Ukonaho, Sophie Reichert, Diogo Dos Santos, U Kyaw Nyein, Win Htut, Virpi Lummaa

**Affiliations:** 1Department of Biology, University of Turku, FIN-20014 Turku, Finland; susanna.s.ukonaho@utu.fi (S.U.); reichert.sophie@gmail.com (S.R.); diogo.francodossantos@utu.fi (D.D.S.); virpi.lummaa@gmail.com (V.L.); 2Myanma Timber Enterprise, Ministry of Natural Resources and Environmental Conservation, Gyogone Forest Compound, Bayint Naung Road, Insein Township, Yangon, Myanmar; kyaw.nyein.mte@gmail.com (U.K.N.); winhtut641@gmail.com (W.H.)

**Keywords:** *Elephas maximus*, cortisol, faeces, heterophils, lymphocytes, welfare

## Abstract

**Simple Summary:**

Animals are kept in captivity for various reasons worldwide. Throughout its range countries, the Asian elephant is used for various purposes, with a significant proportion of the remaining population working as draft and transport animals in the timber industry. However, captivity can also lead to compromises in welfare that need to be quantified for successful intervention. A key way of assessing an animal’s well-being in wildlife and zoo biology is to measure its stress. Previous studies have found positive, negative, or no relationship between two commonly used measures of stress: stress hormones and the ratio of two types of white blood cells—heterophils to lymphocytes. Our study is one of the first to show a positive and consistent link between these two measures in semi-captive Asian elephants from Myanmar, irrespective of sex, age, or environmental context. Our results show that using the heterophil/lymphocyte ratio from blood smears on-site may offer a potentially cheaper and faster way to determine stress than measuring faecal glucocorticoid metabolite concentrations in the laboratory.

**Abstract:**

Animals are kept in captivity for various reasons, but species with a slower pace of life may adapt to captive environments less easily, leading to welfare concerns and the need to assess stress reliably in order to develop effective interventions. Our aim was to assess welfare of semi-captive timber elephants from Myanmar by investigating the relationship between two physiological markers of stress commonly used as proxies for welfare, faecal glucocorticoid metabolite concentrations (FGM) and heterophil/lymphocyte ratios (H/L), and link these measures to changes in body condition (determined by body weight). We further assessed how robustly these two markers of stress performed in animals of different age or sex, or in different ecological contexts. We measured FGM concentrations and H/L ratios between 2016 and 2018 from 316 samples of 75 females and 49 males ranging in age from 4 to 68. We found a positive and consistent link between FGMs and H/L ratios in Asian elephants, irrespective of their sex, age, or ecological context. Our results will help to inform managers of (semi-) captive elephants about using heterophil/lymphocyte ratio data from blood smears on site as a potentially cheaper and faster alternative to determining stress than measuring faecal glucocorticoid metabolite concentrations in the laboratory.

## 1. Introduction

Animals worldwide are kept in captivity for diverse reasons: in zoos for entertainment, education and research; as pets; in research facilities for medical, pharmaceutical, or product testing; as livestock for dairy, meat, and other animal products; or as working animals for draft, transport, rescue, or law enforcement. When animals are transferred from their natural wild environment into such a setting of captivity, the original selection pressures experienced in nature from environmental constraints, limited resources, competition, and predation often disappear (e.g., [1]). Such differences between wild and captive environments can mean that animals brought to captivity might not be well-adapted to their new environment, thus leading to welfare concerns. For example, factors such as lack of exercise, novel group compositions, and social interactions (or the lack of them) can increase stress in captive animals [2]. However, after several generations of living in captivity, at least certain species can become docile and adapted to their new captive environments and show increased longevity and a delayed onset of senescence compared to their wild counterparts [1]. When transferred into a captive environment, some species exhibit a strong change in their behavioural tendencies [3]. It has been suggested that it is probably more difficult to adapt to captive environments by evolutionary means for species with a slower pace of life in comparison to those with a faster pace [1], and for some species this may not be possible at all. A good example is the Asian elephant (*Elephas maximus*), which humans have been using for various purposes for over 4000 years [4]. During this time, Asian elephants have been tamed, but never truly domesticated or selectively bred, since their morphology, behaviour, and life-histories do not allow it.

Assessing the welfare of such species is crucial because of the ethical responsibilities humans have towards captive animals, but the well-being of animals employed in human services (e.g., timber elephants) is also directly linked to economic outcomes. Though it is difficult to find one clear definition of welfare, many ways to assess the well-being and welfare of captive animals have been established and are continuously being developed [5]. Because we currently cannot quantitatively assess what animals “feel”, definitions of welfare range from how healthy an animal is and if it has what it needs [6] to describing the state of an animal and how it copes with its environment [7]. One important method of assessing an animal’s state and well-being in wildlife and zoo biology is to measure its stress [7,8]. The term “stress” can be used to (a) describe a harmful external stimulus that the animal experiences, or (b) the physiological and behavioural reactions of the animal to cope with the stimulus, and (c) the condition when an animal is exposed to a harmful stimulus over a longer time period, which can lead to disease (reviewed in [9]). In general though, stress can be defined as a threat to homeostasis, and an individual can experience stress as a result from real threats (“acute physical crisis” e.g., predator attack) to perceived threats (“sustained psychological stress” e.g., predator cues) [10]. Furthermore, stress can occur when suffering from a disease, when confronted with dominant conspecifics (social stress), or when experiencing extreme environmental situations that threaten homeostasis [11,12,13]. In addition, various internal and external factors can affect stress or stress responsiveness in animals, such as age [14,15], sex [16,17,18], and ecological conditions [19,20,21], or even positive challenges such as breeding interactions, and interpreting stress hormone concentrations is therefore not always straightforward [22,23] and can be context-dependent [24]. Nevertheless, how well an animal can cope with threatening situations is generally reflected by its stress response and therefore, measuring stress can be a useful biomarker to assess the health or well-being of animals.

A common way to assess the effects of stressful conditions is to measure glucocorticoid (GC) hormone concentrations in blood samples or GC metabolites in faecal samples (e.g., [25,26]). The release of GC hormones is governed by the hypothalamo–pituitary–adrenocortical (HPA) axis and is triggered when individuals are confronted with a stressor. The HPA axis then orchestrates a series of physiological and behavioural reactions that help the animal to cope with the stressor. Studies investigating baseline and stress-induced GC concentrations often use blood serum or plasma samples [27,28,29]. However, to accurately measure baseline GC concentrations, blood samples have to be collected within 3 min after capture [30], and a strict capture protocol is therefore required to ensure reliable assessment of stress-induced GC concentrations [31]. This is not always possible, especially with studies on large mammals in natural conditions. Alternatively, faecal cortisol/corticosterone or glucocorticoid metabolites (FGM) concentrations can be measured, which reflect a more integrated level of circulating glucocorticoids and therefore an individual’s stress exposure over a longer time [32].

More recently, ecologists have begun to assess stress with the ratio of two different types of white blood cells: heterophils to lymphocytes (H/L ratio) [33]. Heterophils (or neutrophils, depending on the species) and lymphocytes are both involved in immunological processes and constitute almost 80% of all white blood cells in mammals. In Asian elephants, monocytes are more abundant compared to other mammals, with lymphocytes and heterophils constituting 56% to 67% of all white blood cells, depending on the study [34,35]. While heterophils are phagocytic white blood cells, lymphocytes are responsible amongst others for immunoglobulin production and immune function (see [33] and references therein), and the number of both cell types can be easily assessed through cell counts of blood smears. Heterophil and lymphocyte numbers both react towards stress, but in different directions. While heterophil numbers increase during a stressful event, lymphocyte numbers decrease and previous evidence suggests that an increase in corticosteroids can drive the changes in heterophils and lymphocytes numbers (reviewed in [33]). The original studies on stress and the H/L ratio were performed in domesticated fowl (e.g., [36]), and more recent studies have shown the relationship between stress and haematological measures also in mammals and wild birds [37,38,39]. Although cortisol and N/L ratio (N for neutrophils, which are also used for this measure in mammals) increase in parallel e.g., in cattle after dehorning [40], in pigs after transport [41] or in several primate species in captivity (e.g., [42]), consideration has to be given to factors that influence these ratio values, e.g., age or different life history stages [33,36].

In the present study, we investigate variations in two markers of stress—FGM concentrations and H/L ratios—for the first time in Asian elephants and link these measures to changes in body condition (determined by body weight). Body condition (or body weight as a proxy) is often used as a physical marker of welfare, and it has been suggested that FGM concentrations can be a valuable tool to monitor body condition or health [43]. Our aim was therefore to test if our two markers of stress would be linked to body weight in Asian elephants as well. Our study individuals are semi-captive Asian elephants working in the timber industry in Myanmar. These elephants are used as draft animals to remove logs from the forest and constitute an essential part of the timber economy in Myanmar, an important contributor to the Myanmar GDP. However, the sustainability of the population is uncertain [44], and assessing stress levels in this semi-captive population will contribute to improved vital rates, better productivity, and developing better ways to increase welfare. In contrast to fully captive individuals e.g., in zoos, the timber elephants can forage freely at night and express a large range of their natural behaviours and social interactions in their natural habitat [45], hence the designation ‘semi-captive’. Our study can therefore also bring new insights on the stress physiology of Asian elephants in general, as they are more comparable to wild elephants than fully captive elephants. Importantly, our previous work on this population has already established that in general, males display higher mortality than females [46], the overall health of the animals changes vastly across ages [47], and the vital rates and health measures show strong variation across the three different climatic and working-life seasons in Myanmar [19].

The aims of this study are therefore to assess wellbeing in semi-captive Asian elephants by (1) assessing variation in FGM concentrations; (2) variation in H/L ratios; (3) investigating the relationship between FGM concentrations and H/L ratios in males and females, for different age categories, and for different seasons of the year; and (4) to study the relationship between FGM concentrations and H/L ratios with body weight. It should be noted that heterophils occur in birds and reptiles (and therefore the H/L ratio is usually used), while mammals have neutrophils (and thus the N/L ratio is used). However, like the other members of *Afrotheria*, elephants have heterophils instead of neutrophils and therefore the H/L ratio is used in the current study. Though they can structurally differ, heterophils and neutrophils fulfil the same immunological functions.

## 2. Materials and Methods

### 2.1. Study Population

About 5000 semi-captive elephants [45] are living in the Republic of the Union of Myanmar, and over half of the semi-captive population is state-owned through Myanma Timber Enterprise (MTE). MTE maintains detailed log-books on every individual elephant, which enables the recording of data for all registered elephants. These include registration number and name, birth origin (wild-captured or captive-born), date and place of birth (date estimated for wild-captured elephants), year and place of capture (if wild-captured), and sex. The age of captive-born elephants is known precisely, while wild-caught elephants are aged by comparing their height and a range of physical features at capture with captive elephants of known age. For decades, around half of all timber elephants in the population were captured from the wild and tamed to work, but capture rates have reduced since 1997, and consequently the proportion of newly captured wild elephants working in timber camps has declined [45]. During day-time, the elephants are employed in the timber industry as transport and draft animals, yet due to the semi-captive keeping system mortality rates [48], reproductive profiles [49] and social behaviours [50] resemble those of wild elephants. Work during state-set working hours is carried out under the guidance of a mahout in working groups of six elephants, with a head mahout managing group operations. At night, the elephants forage unsupervised in nearby forests where they mate both with each other and wild elephants. The working tasks of the timber elephants vary according to their life stage. Until taming at age four or five [45,46], the calves are kept with their mother and allomothers who nurse the calves. After taming, the elephants are trained to follow command given by the mahouts and to carry out different work tasks. Until the age of around 20 years, juvenile elephants are used as transport animals for light duties. At around 20 years age, the elephants also start to work as draft animals, pulling and handling logs and transporting them out of the forest [45], until retirement at around age 54. Consequently, in all our analyses, age was included as a four-level factor (taming 4–10 y.o., training 10–20 y.o., working 20–54 y.o., retired 54– y.o.). The Myanmar Government and the Ethics Committee of the University of Turku approved this research.

### 2.2. Study Outline

Our study consists of elephants born 1948–2012. This study included 124 elephants (75 females and 49 males) ranging in age from 4 to 68 years (females: 5–68, median: 12; males: 4–65, median: 9) from Kawlin and Katha logging agencies (both in Sagaing Division). Of the 124 elephants, 18 elephants were wild-captured and 106 captive-born. The combined dataset of FGM concentrations and H/L ratio samples consisted of 316 observations from 124 individuals in total, where 144 samples (49 individuals) and 172 samples (75 individuals) were from males and females, respectively, 140 (40 individuals) from taming calves, 110 (45 individuals) from training calves, 46 (32 individuals) from working adults, and 20 (13 individuals) from retired adults. All data were collected across Myanmar’s distinct three seasons (hot season: March–May, monsoon season: June–September, and cold season: October–February), so seasonal variation in FGM concentrations and H/L ratios could be accounted for. Data on individual elephant FGM concentrations and H/L ratios were collected within the same season given that both stress markers reflect long-term rather than immediate responses to stress, although the vast majority of these two measures were obtained within days (median difference 1 day). Our dataset included 1 pregnant female and 15 lactating females with calves at heel under 4 years old at the time of sampling. None of the elephants involved in this study suffered from severe health problems due to infection or disease.

### 2.3. Physiological Markers of Stress

#### 2.3.1. Faecal Sample Collection and FGM Analysis

To monitor FGM concentrations in the timber elephants, from 2016 to 2018, 316 faecal samples from 124 animals were collected as soon as possible after defecation in the morning and were stored in a ziplock bags at −20 °C until drying in a hot air oven at 50 °C. Dried samples were shipped for further analysis to the Veterinary Diagnostic Laboratory, Chiang Mai University, Thailand. Glucocorticoid metabolites were extracted from faecal samples using a protocol for boiling extraction, and glucocorticoid concentrations were measured by enzyme immunoassay (EIA) for glucocorticoid metabolites using a polyclonal rabbit CJM006 antibody [51]. To achieve consistent results, the original EIA protocol was optimised by using Nunc MaxiSorp^®^ plates, room temperature substrate reagents and dark incubation conditions. FGM concentrations varied between 19.59 and 195.8 ng/g/faeces, a range that has been observed before for Asian elephants although comparisons between different studies should always be interpreted with caution. Intra-assay variation was 11.8%, inter-assay variation was lower than 10%, and the minimum detection limit was 0.08 ng/g.

#### 2.3.2. Blood Smear Preparation and Heterophils/Lymphocytes Ratio

To assess heterophil/lymphocyte ratio, 316 blood samples were obtained from an ear vein of 124 elephants during three occasions (March/April, July, November) in 2016–2017 and once (March/April) in 2018 in the mornings after collecting the animals from the forest where they foraged freely during the night. Samples were refrigerated for a maximum of 24 h before analysis in the laboratory. Blood samples collected in EDTA tubes were used for determining the leucocyte count (for H/L ratio), where the blood cells were counted manually using a blood smear stained with Romanowsky stain. The ratios were calculated based on the amounts of heterophils and lymphocytes in the blood sample slide. H/L ratios in the sampled individuals varied from 0.30 to 2.8, which is considered within a normal range for elephants, although the normal parameters for free-ranging animals are still unknown.

### 2.4. Body Weight

To investigate how our two correlates of stress co-varied with between- and within-individual variation in body condition, body weight data (as a proxy for condition) from 2016 to 2018 was collected simultaneously with the blood and faecal sample collection using an Eziweigh 3000^®^ scale, capable of weighing up to 9000 kg to the nearest 10 kg. To control the effect of age on elephant’s size, and following previous work (e.g., [52,53]), we calculated and used standardised weights as dependent variables (on average two measurements per individual with a range of one to eight measurements per individual), so that we could measure the changes in body weight relative to the typical weight at a given age. These standardised weights were obtained by dividing an individual’s measured weight by the predicted weight at that age, as obtained from von Bertalanffy growth curves from this population (presented in [53]). We used separate curves for males and females, and the final dataset contained 198 observations from 113 individuals.

### 2.5. Statistical Analysis

All statistical analyses were performed with the software R (version: 3.4.4, [54]), using the package nlme. For all linear mixed models, the distributions of FGM concentration and H/L ratio and standardised body weights were checked and confirmed for normality using residual plots. To analyse the relationship between FGM concentration and H/L ratio, we constructed linear mixed-effects models, with FGM concentration as the dependent variable, H/L ratio as the independent variable and elephant identity as a random term to account for repeated measures. To investigate potential differences in the relationship between FGM concentration and H/L ratio for males and females, for different age classes, and for seasons, we tested for interactions between the H/L ratio and the variables of interest (sex, age class, and season) by fitting these interactions one at a time alongside the main effect of sex (two levels: males and females), age class (four levels, see above) and season (three levels, see above) into the original model. All models included sex, age class, season, and origin as confounding variables as well, given that our previous studies have shown that these are linked with differences in health, survival and reproduction of the elephants [19,46,47]. To investigate the effects of FGM concentration or H/L ratio on body weight, we built two linear mixed-effects models, controlling for sex, age class, origin, and season as above, and including elephant identity as a random effect to account for repeated measures. The residuals of all LMMs adhered to the assumption of normality.

## 3. Results

### 3.1. Minima, Maxima and Averages of FGM Concentrations and H/L Ratios

Both FGM concentrations and H/L ratios varied widely between and within individuals with a mean and SD of 74.9 ± 28.3 and 1.10 ± 0.525, respectively. FGM concentrations ranged from 19.6 ng/g to 195.8 ng/g, and H/L ratios ranged from 0.30 to 2.8. The highest FGM concentration was found in a male taming elephant during hot season, whereas the lowest FGM concentration was identified in a female retired elephant during hot season. The highest H/L ratio was found in a female training elephant during cold season, whereas the lowest H/L ratio was identified in a female training elephant during hot season. The minimum, maximum, and average values for FGM concentrations and H/L rations for semi-captive Asian elephants of our study population can be found in Table 1.

### 3.2. The General Correlation between H/L Ratio and FGM Concentrations

Overall, in our study population of semi-captive Asian elephants, higher FGM concentrations were statistically significantly associated with higher H/L ratios (LMM: *b* = 6.23, F_1,186_ = 6.06, *p* < 0.01; Figure 1a,b, Table 2). Elephants with an increase in H/L ratio from 1 to 2 showed a 8.9% increase in FGM concentrations.

### 3.3. Correlations between H/L Ratio and FGM Concentrations Depending on Sex

Elephant males showed on average 11% higher FGM concentrations compared to females (LMM: *b* = 7.45, F_1,121_ = 8.38, *p* < 0.01). However, the correlation between FGM concentrations and H/L ratios remained robust for both sexes, despite such overall sex differences in FGM concentrations, as indicated by a non-significant interaction between sex and H/L ratio (LMM: *b* = −4.52, F_1,185_ = 0.570, *p* = 0.451, Table 2).

### 3.4. Correlations between H/L Ratio and FGM Depending on Age Classes

FGM concentration was not significantly associated with age class in our sample (F_3,186_ = 0.163, *p* = 0.921), and the correlation between FGM concentrations and H/L ratios remained robust for all age categories, as indicated by a non-significant interaction between age category and H/L ratio (LMM: F_3,183_ = 0.817, *p* = 0.486, Table 2).

### 3.5. Correlations between H/L Ratio and FGM Concentrations Depending on Season

When considering seasonal effects, season was linked to FGM concentrations (F_2,186_ = 15.0, *p* < 0.0001), with elephants showing 16% lower levels of FGM in the hot season (LMM: *b* = −11.5, df = 188, *p* < 0.01) and 17% higher levels of FGM during the monsoon than in the cold season (LMM: *b* = 12.1, df = 188, *p* = 0.0183). Despite such large overall variation in FGM across the different seasons, the correlation between FGM concentrations and H/L ratios remained robust throughout the seasons, as indicated by a non-significant interaction between season and H/L ratio (F_2,184_ = 0.246, *p* = 0.782, Table 2).

### 3.6. FGM Concentrations and H/L Ratio and Their Links to Body Weight

The observed body weights ranged from 100 kg to 2360 kg across all ages and sexes. When controlling for sex (*p* > 0.05), season (*p* > 0.05), origin (*p* > 0.05), and age category (*p* < 0.001), the H/L ratio was not statistically significantly related to body weight (LMM: *b* = −0.021, df = 183, *p* = 0.343). However, when controlling for sex (*p* > 0.05), season (*p* > 0.05), origin (*p* > 0.05), and age category (*p* > 0.05), higher FGM concentrations were linked to lower body weight (LMM: *b* = −0.00185, df = 81, *p* < 0.05, Figure 2). For example, an increase in FGM concentration from 50 to 100 ng/g/faeces corresponds to on average 20% lower than expected body weight, which translates into a 400 kg weight loss in a 2000 kg adult.

## 4. Discussion

An effective way of assessing an animal’s well-being in wildlife and zoo biology is to measure its stress or stress response [7,8]. Across species, previous studies have usually found positive relationships between two commonly used markers of physiological stress: GC concentrations and H/L ratios (reviewed in [33]). However, in other studies, this relationship is sometimes weak or not existing [37,55]. Our study is one of the first to establish the link between variation in faecal glucocorticoid metabolite concentrations and the H/L ratios in Asian elephants, showing consistent correlations in both sexes, in animals of different life stages, and in different ecological contexts (here, season). We also show the correlation of the two physiological measures of stress with an indicator of physical welfare, body weight, on a large longitudinal sampling of semi-captive timber elephants in Myanmar. Our results could help to inform handlers and managers of semi-captive (as well as captive) elephants about using H/L ratio data as a potentially faster and cheaper way to assess stress in their animals, which can be easily combined with rapid assessment of physical measures of welfare related to physiological stress.

Elevated FGM concentrations and H/L ratios can be a sign of chronic stress and lower health or wellbeing in humans and non-human animals [56,57,58,59], and individuals expressing higher H/L ratios are also considered to be more susceptible for diseases in the future [60]. In general, GCs as well as H/L ratios are also known to predict survival [61,62] and are therefore key indicators of welfare. We found that elephants with higher FGM concentrations also showed higher H/L ratios, consistent with the assumption that changes in FGM concentrations and H/L ratios do not occur as quick (days for FGM concentrations, hours for H/L ratios) as changes in GC concentrations in the blood stream [58] and therefore they both reflect longer-term effects on measures of stress. Alternatively, other studies (mainly in birds) suggest that baseline GC concentrations from blood plasma levels and the H/L ratio do not indicate the same type of stress [37], since baseline and stress-induced GC concentrations are more sensitive to severe stressors than the H/L ratio, which already increases in response to subtle stressors (see also [58]). However, the integrative nature of FGM concentrations, which reflect GCs excreted over a longer time period, are probably more likely to be related to H/L ratios, since changes in the H/L ratio can also reflect long-term changes in the environment rather than short-term severe stressors [58]. A study investigating welfare indicators in laying hens experiencing a change in their environment found that H/L ratios were not correlated with plasma corticosterone concentrations, though the authors found a strong positive trend between H/L ratios and FGM (here corticosterone) concentrations [63]. Since our study is correlative in nature, we cannot say if FGM affects H/L ratios or the other way around in elephants. However, it has been shown that GCs in general can have multiple effects on the immune system [33,64], specifically the relative numbers of different types of leucocytes. Various sources of social and physical stress can influence the H/L ratio, and the H/L ratio has, therefore, been suggested to be a valuable measure of stress in vertebrates [33,36]. Our findings of a positive relationship between FGM concentrations and the H/L ratio corroborate these assumptions. A potential drawback of our dataset is that we actually cannot say for sure which individuals suffer most from chronic stress. Range values of FGM concentrations and H/L ratios often do not exist for unstressed free-living animals in general [55], nor for unstressed Asian elephants in particular. In addition, a recent review of the chronic stress literature has shown that a general “endocrine profile” of a chronically stressed wild animal does probably not exist [65]. However, considering that our data show a 10-fold difference between the minimum and maximum values for both FGM concentrations and H/L ratios, we do see high variation in both measures of long-term stress. This could mean that elephants with high FGM concentrations and H/L ratios suffer more from chronic stress than others. In addition, considering the predictive power of FGM concentrations and H/L ratios with regards to overall health and fitness [22,58,59,60,62], our results can be valuable for estimating which elephants are more prone to stress-related health risks than others and allow targeting of those individuals to find out why they are more afflicted and to mitigate potential stressors. Therefore, assessing FGM concentrations and H/L ratios are two valuable, reliable, and complementary methods of estimating elephant welfare and health.

Another key result we discovered is that the correlation between the two measures of stress was robust in both sexes, despite males having slightly higher FGM concentrations than females. In this population, males also display different behavioural patterns [66], have a higher overall mortality [46], and a higher parasite-associated mortality [67] across all ages. In general, this is not surprising: Sex can affect stress and stress responses [16,17]. Though a previous study on FGM concentrations and seasonality found no differences between the sexes [19], the weak difference we found could be due to a higher sample size. In general, male and female elephants are employed for the same work purposes, experience the same keeping system and environment, and live in mixed sex groups; there are no obvious explanations as to why males would have higher FGM concentrations than females in our study population, although it is possible that males are used for more demanding work tasks than females due to their (typically) larger body size, or that their training is more harsh than that of females. A study on Asian elephants living in tourist camps in India actually found the opposite, with females having higher FGM concentrations than males [68]. However, the same study found that in zoo elephants, males had higher FGM concentrations than females [68]. Higher stress or GC in general in female Asian elephants could be due to them being pregnant or lactating and needing more daily energy. Since male Asian elephants usually live solitary or in loose bachelor groups in the wild [4], living in logging camps with several same-sex individuals could be more stressful for males than females, who are naturally living in larger family groups [4]. In African elephants (*Loxodonta africana*), the presence of fertile females seems to be a stressor for males [69]. Slight differences in FGM concentrations between the sexes could however also be simply due to differences in how males and females metabolize steroid hormones [70].

We also found that FGM concentrations did not vary between different age classes, but that the correlation between FGM concentrations and the H/L ratio was similar whatever the age of the animal. The finding that different age classes did not differ in FGM concentrations is surprising, as we expected age-dependent differences in this marker of stress in elephants. In our study population, there are clear differences in reproduction and mortality between different age categories [46,49], but, despite these findings, FGM concentrations in general did not differ significantly between elephants of different ages classes included in our study. One reason might be that in this study we compared FGMs across four broad age categories instead of investigating fine-tuned patterns across all ages. Age-dependent differences in GCs have been found in other mammals, e.g., free-ranging younger male giraffe (*Giraffa camelorpadalis giraffa*) bulls had higher FGM concentrations when compared to older age classes [14], while older wild grey mouse lemurs (*Microcebus murinus*) showed higher FGM concentrations [20]. It should be noted that those effects were dependent on social context and season [14,20] and that interpretations of FGM concentrations are often highly context-dependent [71,72]. A study on Asian elephants from tourist camps from Thailand found a strong trend between age and FGM concentrations, with older elephants having lower FGM concentrations [73]. However, there is no consistent evidence that this relationship always holds up in elephants, since other studies found no relationship between age and FGM concentrations (e.g., [74,75]). The fact that there was no significant interaction between the H/L ratio and age class, however, indicates that the relationship between FGM concentrations and the H/L ratio is robust in elephants of all age categories, and therefore both variables are valuable markers of stress and welfare in Asian elephants.

We also tested how the two stress measures performed in different ecological environments. Various seasonal effects on GC concentrations have been demonstrated in mammalian and bird species [20,76,77], and seasonal effects on FGM concentrations have been shown before in our study population [19]. Elephants in Myanmar experience three very distinctive seasons that vary in temperature (high during hot season and monsoon and low during cold season), food abundance (high during monsoon, low during hot season), disease prevalence (peaks during monsoon, often due to flooding), and workload (no work during hot season, working during monsoon, and cold seasons) [19,45,67]. Elephants showed the lowest FGM concentrations during the hot season and highest FGM concentrations during the monsoon season [19]. Elephants rest during the hot season and do not participate in any work, spending the days in the forests foraging. During monsoon season, elephant FGM concentrations could increase due to increased workloads and a high temperature-humidity index, meaning elephants have to invest more energy for thermoregulation [19]. The fact that we find a consistent relationship between FGM concentrations and H/L ratios in all seasons, despite the timber elephants experiencing drastically different environments and work schedules between the different seasons, again lends credence that both markers can be used to assess stress in Asian elephants in a wide range of settings and contexts.

Our finding that one marker of stress (FGM concentrations) is linked to body weight (as a proxy for body condition) but not the other (H/L ratio) is somewhat surprising, since we expected both of these markers of long-term stress to be negatively correlated to body condition. However, at least the link between FGM concentrations and body weight is not unexpected, since the relationship between GCs and body condition has been shown several times before in other species [43,78]. GCs are involved not only in an animals’ stress response, but also in energy mobilisation, fat metabolism, and muscle catabolism, and these factors can negatively influence body condition. However, it has been noted that the cause and effect of the relationship between GCs and body condition remains unclear [79], and the dynamics of this relationship may differ between species and context [71,78]. A previous study on the same population of Asian elephants has actually shown a positive relation between FGM concentrations and body weight [19], though the related body condition was measured 3 months after measuring FGM concentrations. Considering the multiple effects of GCs on the immune system and the H/L ratio, and that elephant body weight can vary largely between individuals of the same age, perhaps it is then less surprising that we could not detect a direct link between the H/L ratio and body weight. Finding a link between a physical marker of welfare (body weight) and at least one physiological marker of welfare indicates that our physiological markers can be useful when assessing welfare and health in Asian elephants.

To conclude, Asian elephants are long-lived highly-social mammals with the potential capacity of self-recognition [80,81], and assessing their welfare is an ethical obligation. It has been noted that the relationship between different measures of welfare is multifaceted and not always straight-forward [82], and therefore care always needs to be taken when interpreting multiple welfare measures. While this is certainly true, our study has shown that two common physiological measures of stress (FGM concentrations and H/L ratio) are correlated in the same way in both sexes, in different age groups, and in highly contrasting ecological settings. Many factors, of which some we have addressed here, such as age, life-history stage, seasonality, human handling, between individual differences, and heritability, can influence the H/L ratio (reviewed in [58]). Other factors affecting FGM concentrations and H/L ratios that were not considered in this study are social factors, diseases, and parasite load or reproductive state. These factors, in combination with sex, age, and season, can have complex interactive effects on measures of stress and welfare. Therefore, one should always interpret measures of FGM concentrations and H/L ratios carefully and in the specific contexts they were measured. Considering the consistent link between FGM concentrations and the H/L ratio in our population of Asian elephants, irrespective of sex, age category or season, we can assume that both stress measures can be used to assess the status and welfare of individual elephants. Hence, to assess stress in elephants, measuring the H/L ratio from blood smears on site can be a cheaper and faster alternative to determining FGM concentrations in the laboratory via more expensive, slower immune assays. The identification of different white blood cells from blood smears is, however, not always straight-forward, and therefore training of local veterinarians and staff will be necessary if H/L ratios are to be used more widely as a measure of stress.

## Figures and Tables

**Figure 1 animals-10-00094-f001:**
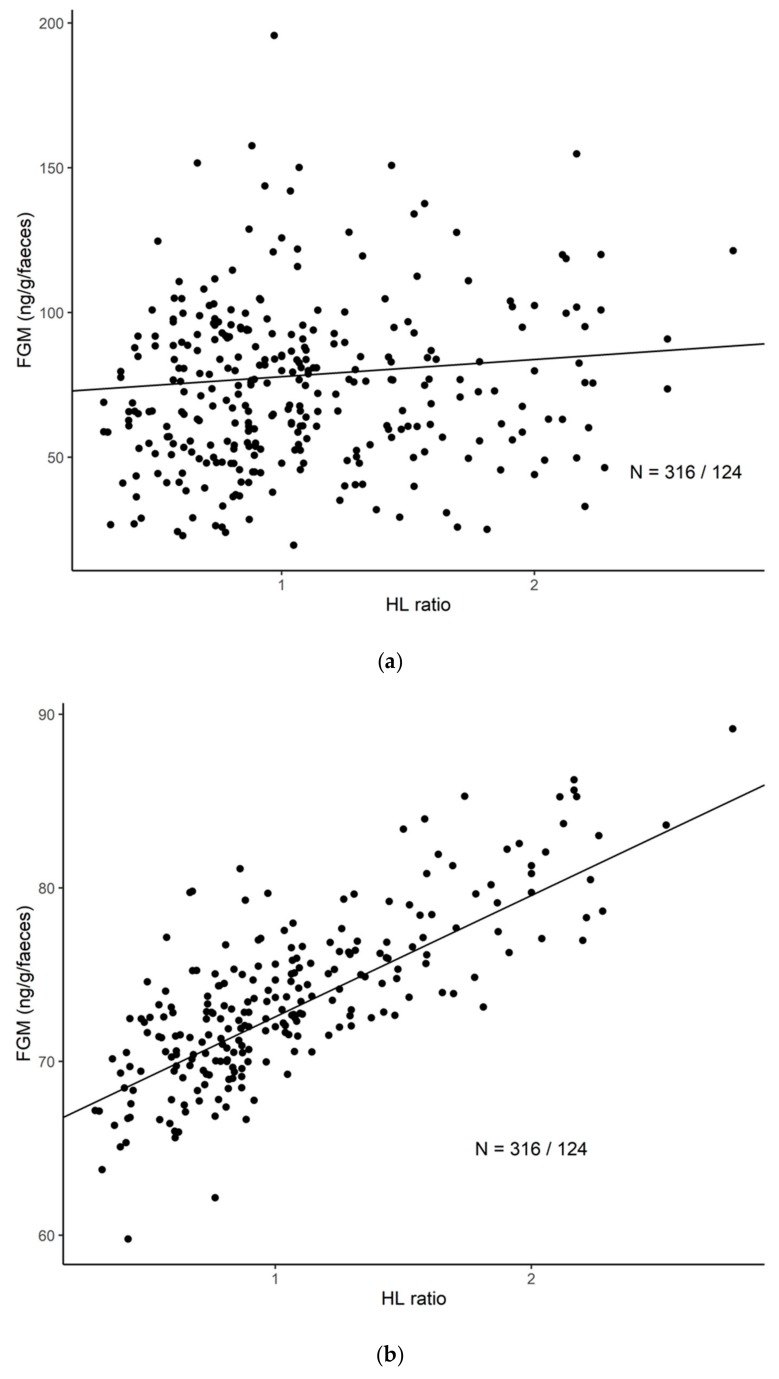
(**a**) The relationship between the H/L ratio and FGM concentrations in Asian elephants. The predicted curve is plotted against the observed raw data. The sample size shown refers to the number of samples/total number of individuals included in the study. (**b**) The relationship between the H/L ratio and FGM concentrations in Asian elephants. Here, the predicted curve is plotted against the predicted data after adjusting for the confounding variables shown for Model 1 in Table 2. The sample size shown refers to the number of samples/total number of individuals included in the study.

**Figure 2 animals-10-00094-f002:**
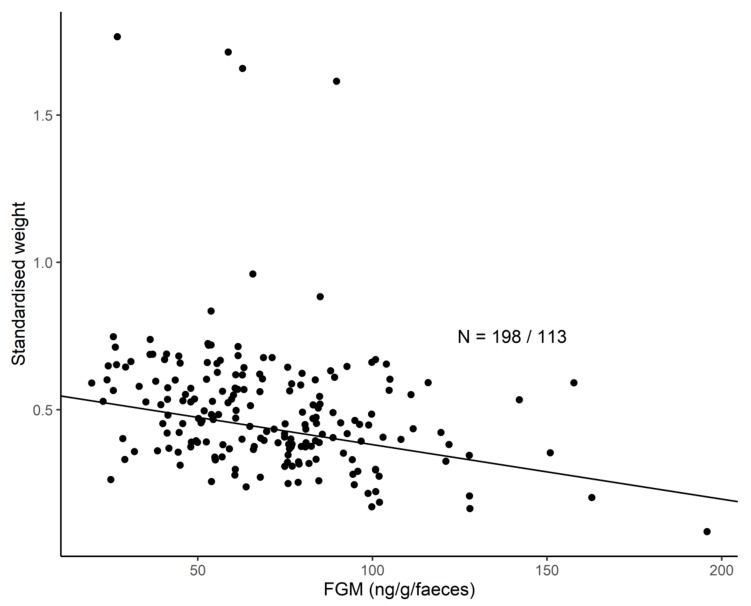
The relationship between FGM concentrations and standardised body weight. The predicted curve is plotted against the predicted data after adjusting for the confounding variables shown for Model 5 in Table 2. The sample size shown refers to the number of samples/total number of individuals included in the study.

**Table 1 animals-10-00094-t001:** Minima, maxima, and average values of FGM concentrations and H/L ratios for different categories of Asian elephants (sex, age, and season).

FGM Concentration (ng/g)/H/L Ratio				
		Minimum	Average	Maximum
Sex	male	24.0/0.363	78.9/1.05	195/2.79
	female	19.6/0.296	71.5/1.08	162/2.53
Age	taming	24.30/0.363	80.0/1.05	195/2.79
	training	22.9/0.296	73.9/1.09	162/2.28
	working	25.8/0.323	65.6/1.01	142/2.23
	retired	19.6/0.371	63.3/1.23	143/2.53
Season	hot	19.6/0.296	66.8/0.976	195/2.53
	monsoon	44.9/0.488	91.6/0.975	158/2.17
	cold	24.3/0.425	81.5/1.31	150/2.79

**Table 2 animals-10-00094-t002:** Results from linear mixed-effects models for FGM concentrations and body weight as dependent variables: dependent variable (parameter), independent and confounding variables (variable, confounding variables in italics), interactions (asterisk), F-value (F), degrees of freedom (DF), and *p*-value. The statistics of the main effects reported in the results section are from Model 1 (for a residual QQ plot indicating the model fit of Model 1 see Appendix A).

Parameter	Variable	F	DF	*p*-Value
Model 1				
FCM	H/L	6.06	1, 186	0.0147
	sex	8.38	1, 121	<0.01
	*origin*	13.5	1, 121	<0.001
	season	15.0	2, 186	<0.0001
	age	0.163	3, 186	0.921
Model 2				
FCM	H/L	6.34	1, 185	<0.01
	*sex*	7.92	1, 121	<0.01
	*origin*	13.3	1, 121	<0.001
	*season*	15.0	2, 185	<0.0001
	*age*	0.171	3, 185	0.916
	H/L*sex	0.570	1, 185	0.451
Model 3				
FCM	H/L	6.11	1, 183	<0.01
	*age*	4.28	3, 183	<0.01
	*sex*	3.64	1, 121	0.0586
	*origin*	7.19	1, 121	<0.01
	*season*	14.4	2, 183	<0.0001
	H/L*age	0.817	3, 183	0.486
Model 4				
FCM	H/L	2.32	1, 184	0.130
	*season*	19.9	2, 184	<0.0001
	*origin*	8.28	1, 121	<0.01
	*sex*	7.27	1, 121	<0.01
	*age*	0.162	3, 184	0.922
	H/L*season	0.246	2, 184	0.782
Model 5				
Standardised body weight	FCM	7.14	1, 80	0.00920
	*season*	2.02	2, 80	0.139
	*sex*	0.0927	1, 110	0.761
	*age*	0.876	3, 80	0.457
	*origin*	0.000800	1, 110	0.977

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
