# Peer review of "Faecal Glucocorticoid Metabolites and H/L Ratio Are Related Markers of Stress in Semi-Captive Asian Timber Elephants"

_animals, 2020, doi:10.3390/ani10010094_

Round 1
Reviewer 1 Report
This is an interesting study that combines a commonly used measure of stress in elephants, faecal glucocorticoids, with heterophil:lymphocyte ratio, which has been widely used across taxa, but only minimally in elephants. The sample size is relatively large, and the authors provide a robust dataset to lay the groundwork for the use of H:L as a welfare measure, comparing age, sex, origin, season and body condition. This manuscript will be a useful addition to the field; I suggest just a few minor edits to improve clarity to anyone less familiar with the topic.
Typically neutrophil:lymphocyte ratio is used in mammals and H:L in birds and reptiles, so I think it would be worth adding a sentence or two to the introduction specifically mentioning that elephants have heterophils instead of neutrophils, so that it is clear to any readers that are unfamiliar with this technique that the ratio is equivalent to the examples you mention.
Secondly, throughout the manuscript I recommend changing FCM to FGM. We know that many species produce a combination of cortisol and corticosterone, therefore you may not be measuring only metabolites of cortisol in faeces, so it is more accurate to refer to faecal glucocorticoid metabolites.
Other minor comments by line:
Title: Suggest changing ‘cortisol metabolites’ to ‘glucocorticoid metabolites’, because most species likely produce both glucocorticoids to some extent, therefore you cannot be sure that you are not also measuring metabolites of corticosterone
Lines 12-14 and 44-47: These are complex lists, so would be easier to read with semi-colons to separate items
Lines 15-16: I would specify that this (significant proportion of the remaining population) is in range countries, or in Myanmar specifically
Line 24: suggest ‘on-site’
Keywords: heterophils is misspelled here
Line 49: should read ‘constraints’
Line 54-56: I think this paragraph might flow slightly better if you move the part of this sentence regarding increased longevity and delayed onset of senescence up to just after the first citation of reference [1] – i.e. transferring into captivity removes selection pressures – increased longevity and delayed senescence – then go on to state that other elements of the captive environment could lead to welfare concerns and change in behavioural tendencies.
Line 61: suggest rephrasing to ‘never truly domesticated or selectively bred’ – elephants have been bred in captivity, but not selectively for specific traits
Line 63: should read ‘not least because’
Lines 70-83: Please add a sentence or two here to acknowledge that an adrenal response can occur in response to positive challenges also – for example increased during social or breeding interactions.
Lines 97-99: Because elephant leukocytes are not necessarily typical of other mammals (monocytes are more abundant, unique morphology etc.), this section might be more informative if you discuss the relative proportions of heterophils and lymphocytes in elephants, rather than mammals in general.
Line 102: this should read ‘heterophil and lymphocyte numbers react’ (or heterophils and lymphocytes react)
Line 103: should read heterophil numbers
Line 103-104: for consistency, please change to lymphocyte numbers decrease
Line 108: please explain that although neutrophils are commonly used in mammals, elephants possess heterophils and not neutrophils (just to make it clear that they are functionally equivalent but it is not a choice of one or the other)
Line 115: please remove the unused ()
Line 160: These categories do not match your description – i.e. 18yo start of working. Can these categories be adjusted to reflect when the change in workload occurs?
Section 2.2: it would be useful here to add a sentence about the health status of the elephants included in this dataset – were all individuals considered to be clinically normal at the time of sample collection, or did any have known infectious/inflammatory conditions?
Line 167: should read H/L
Lines 172-175: Faecal GCs can be highly variable day to day, so those samples collected a month apart might not be as accurate as those collected on the same day as the H:L ratio - did you try running your analyses with and without the data points that were not collected on the same day, to ensure that they were not impacting your overall result?
Line 183-184: although the full method has been published, it would be useful to include a brief sentence with the antibody used, for anyone that cannot access the cited paper and would like to replicate your study
Lines 203-207: I like this idea of using standardized weights, but am having difficulty finding the growth curves you mention in the cited reference - could you perhaps consider including these as supplementary material?
Lines 216-221: please add additional information regarding the reference categories used for each categorical fixed effect, and how different models were compared to determine which was the best model (I’m assuming model 1 was considered to be the best for your data since that is the one that you report in the text? Please could you clarify this).
Line 222: should read ‘built’
Line 237: should read ‘ratios’
Line 238: for consistency, should read H/L
Results – please check consistency of decimal places used in reporting statistics
Line 265: please provide the p value here, rather than p<0.05
Line 265: for consistency, please change spelling to faeces
Lines 284-287: I think it would be useful to add a bit more context to this statement that changes in faecal GC metabolites and H:L do not occur quickly – what do you mean by quickly? Faecal GC metabolites can be reflective of events that occurred the day previously, but is the same true for H:L?
Line 307: should read ‘exist’
Line 331: were any of the females included in your study pregnant or lactating? Were any of the bulls in musth? Reproductive state (including stage of the oestrous cycle) can impact GC concentrations, so should be considered as a possible factors in the sex difference seen here.
Line 387: at least one?
Line 661: confounding variables are not in italics
Table 2: It would be useful to see the effect size/coefficient for each variable (perhaps as another column) – we can see whether there is a significant difference in e.g. FCM and H/L, but not the direction. Please also ensure that the number of decimal places used in the reporting of statistics is consistent. It would also be useful to indicate the reference category used so that the tables can be interpreted without needing to refer back to the methods (this information also needs to be added there).
A couple of other suggestions that could be added to the discussion if this technique is to be used more widely as a measure of stress in individual elephants:
1) care should be taken when assessing blood smears due to different forms of segmented monocytes being categorised as lymphocytes. There are some good reference texts out there, so it will be important to provide criteria (and training) if this technique is to be used more widely, to ensure this is a robust and repeatable measure.
2) What about looking at changes within individuals over time? We know CBCs can be highly individual in this species, so from a clinical perspective, we assess changes against each individual’s reference intervals rather than one produced at a species level – is this something that could be done here too – to look at changes in welfare over time in response to management changes etc.?
Author Response
This is an interesting study that combines a commonly used measure of stress in elephants, faecal glucocorticoids, with heterophil:lymphocyte ratio, which has been widely used across taxa, but only minimally in elephants. The sample size is relatively large, and the authors provide a robust dataset to lay the groundwork for the use of H:L as a welfare measure, comparing age, sex, origin, season and body condition. This manuscript will be a useful addition to the field; I suggest just a few minor edits to improve clarity to anyone less familiar with the topic.
A: We thank the reviewer for their kind and positive words.
Typically neutrophil:lymphocyte ratio is used in mammals and H:L in birds and reptiles, so I think it would be worth adding a sentence or two to the introduction specifically mentioning that elephants have heterophils instead of neutrophils, so that it is clear to any readers that are unfamiliar with this technique that the ratio is equivalent to the examples you mention.
A: This is a very good point and we corrected this omission by adding three sentences to the introduction. (ll. 114-119).
Secondly, throughout the manuscript I recommend changing FCM to FGM. We know that many species produce a combination of cortisol and corticosterone, therefore you may not be measuring only metabolites of cortisol in faeces, so it is more accurate to refer to faecal glucocorticoid metabolites.
A: We agree with the reviewer that it is safer to say “glucocorticoids” instead of “cortisol” for the above mentioned reasons and changed “cortisol” to “glucocorticoids” throughout the manuscript where it is appropriate (see also comment 1 from reviewer 3).
Other minor comments by line:
Title: Suggest changing ‘cortisol metabolites’ to ‘glucocorticoid metabolites’, because most species likely produce both glucocorticoids to some extent, therefore you cannot be sure that you are not also measuring metabolites of corticosterone
A: Changed accordingly (l. 2).
Lines 12-14 and 44-47: These are complex lists, so would be easier to read with semi-colons to separate items
A: We removed the list from the simple summary, but changed it accordingly in the introduction (ll. 43-45)
Lines 15-16: I would specify that this (significant proportion of the remaining population) is in range countries, or in Myanmar specifically
A: We added that information (ll. 12-13).
Line 24: suggest ‘on-site’
A: Corrected (l. 22).
Keywords: heterophils is misspelled here
A: Corrected (l. 38).
Line 49: should read ‘constraints’
A: Corrected (l. 47).
Line 54-56: I think this paragraph might flow slightly better if you move the part of this sentence regarding increased longevity and delayed onset of senescence up to just after the first citation of reference [1] – i.e. transferring into captivity removes selection pressures – increased longevity and delayed senescence – then go on to state that other elements of the captive environment could lead to welfare concerns and change in behavioural tendencies.
A: We have restructured these sentences accordingly (ll. 52-54).
Line 61: suggest rephrasing to ‘never truly domesticated or selectively bred’ – elephants have been bred in captivity, but not selectively for specific traits
A: An excellent point. Corrected (l. 60).
Line 63: should read ‘not least because’
A: Corrected (l. 62).
Lines 70-83: Please add a sentence or two here to acknowledge that an adrenal response can occur in response to positive challenges also – for example increased during social or breeding interactions.
A: We agree with the reviewer that the interaction between adrenal activity and (social) behaviour is complex, and that increases in GCs can be related for instance to increased parental care. We try to address this complexity in ll. 80-81 “interpreting stress hormone concentrations is therefore not always straightforward and can be context-dependent”. If the reviewer could be more specific about the link between adrenal responses and positive social interactions and could provide specific examples we are happy to include these here. We did however add “or even positive challenges such as breeding interactions” in line 79 to address this fact.
Lines 97-99: Because elephant leukocytes are not necessarily typical of other mammals (monocytes are more abundant, unique morphology etc.), this section might be more informative if you discuss the relative proportions of heterophils and lymphocytes in elephants, rather than mammals in general.
A: Good idea! We added information on the relative proportions of lymphocytes and heterophils in Asian elephants in this section (ll. 100-102).
Line 102: this should read ‘heterophil and lymphocyte numbers react’ (or heterophils and lymphocytes react)
A: Corrected to the first suggestion (l. 105).
Line 103: should read heterophil numbers
A: Corrected (l. 106).
Line 103-104: for consistency, please change to lymphocyte numbers decrease
A: Corrected (l. 107).
Line 108: please explain that although neutrophils are commonly used in mammals, elephants possess heterophils and not neutrophils (just to make it clear that they are functionally equivalent but it is not a choice of one or the other)
A: We have now added this information to the end of that paragraph (ll. 114-119).
Line 115: please remove the unused ()
A: Removed (l. 123).
Line 160: These categories do not match your description – i.e. 18yo start of working. Can these categories be adjusted to reflect when the change in workload occurs?
A: We apologize for that mistake. We have now corrected this information (ll. 165-166).
Section 2.2: it would be useful here to add a sentence about the health status of the elephants included in this dataset – were all individuals considered to be clinically normal at the time of sample collection, or did any have known infectious/inflammatory conditions?
A: The elephants included in this study are semi-captive and represent a range of ages, sexes, and health conditions as would be the case for any natural population of animals. They were not clinically assessed to the level that we could out-rule ongoing infections or inflammatory conditions, but none suffered from serious health problems at the time of sample collection. We have now included this information on lines 186-187.
Line 167: should read H/L
A: Corrected (l. 176).
Lines 172-175: Faecal GCs can be highly variable day to day, so those samples collected a month apart might not be as accurate as those collected on the same day as the H:L ratio - did you try running your analyses with and without the data points that were not collected on the same day, to ensure that they were not impacting your overall result?
A: This is a very good point and we thank the reviewer for bringing this up! We checked our dataset and found that there had been an error for matching the FGM measurements and H/L ratios for some individuals. We removed those instances (31 observations) and re-analysed our data, and removing those individuals did not affect our overall results (see new statistics). The reviewer is correct that there are within day fluctuations in plasma GC levels, but they follow a general pattern, and are usually similar for every day. These patterns are also reflected in FGM concentrations. Faecal samples are collected around the same time of day (mornings). We matched our data by the same season, but technically this means that the temporal difference between the two measurements is still very small (median difference between the sampling dates was 1 day). This is because e.g. for some elephants faecal samples were collected in end of March and blood samples were taken in the beginning of April etc. We have now clarified this and included the median difference in sampling dates into the revised manuscript in lines 184-185.
Line 183-184: although the full method has been published, it would be useful to include a brief sentence with the antibody used, for anyone that cannot access the cited paper and would like to replicate your study
A: This information has been added (l. 196).
Lines 203-207: I like this idea of using standardized weights, but am having difficulty finding the growth curves you mention in the cited reference - could you perhaps consider including these as supplementary material?
A: We apologize for this inconvenience and our sloppy mistake. We originally only cited a study where the findings of the original growth curve paper were applied. We now added the original paper that includes the growth curves as well (l. 218).
Lines 216-221: please add additional information regarding the reference categories used for each categorical fixed effect, and how different models were compared to determine which was the best model (I’m assuming model 1 was considered to be the best for your data since that is the one that you report in the text? Please could you clarify this).
A: We did not perform model comparisons per se as the main focus of the study was to test the specific hypothesis that the relationship between FGM measurements and H/L ratios could differ for animals of different sex, age class, and in different seasons, so that we know if they both predict each other depending on a variety of conditions. The confounding variables have been chosen based on previous studies on our population, in which it has been shown that sex, age class, origin, and season have effects on many different morphological, physiological and life-history traits (this information is now added, see lines 236-237). Regarding the comment about reference categories, please see our response below.
Line 222: should read ‘built’
A: Corrected (L. 238).
Line 237: should read ‘ratios’
A: Corrected (L. 259).
Line 238: for consistency, should read H/L
A: Corrected (L. 260).
Results – please check consistency of decimal places used in reporting statistics
A: Thank you for your comment, we present our new results in three significant digits, except for p-values smaller than 0.01.
Line 265: please provide the p value here, rather than p<0.05
A: Corrected (see updated results section).
Line 265: for consistency, please change spelling to faeces
A: Corrected (l. 303).
Lines 284-287: I think it would be useful to add a bit more context to this statement that changes in faecal GC metabolites and H:L do not occur quickly – what do you mean by quickly? Faecal GC metabolites can be reflective of events that occurred the day previously, but is the same true for H:L?
A: We agree, and we now provide more information. The sentence now reads “…do not occur as quick (days for FGM concentrations, hours for H/L ratios) as changes in of GC concentrations in the blood stream…” (ll. 332-333).
Line 307: should read ‘exist’
A: Corrected (L. 354).
Line 331: were any of the females included in your study pregnant or lactating? Were any of the bulls in musth? Reproductive state (including stage of the oestrous cycle) can impact GC concentrations, so should be considered as a possible factors in the sex difference seen here.
A: This is a very good comment. Reproductive state (females pregnant, bulls in musth) can indeed affect GCs and FGMs. No bulls in our sample were in musth, since it would be too dangerous to sample those individuals and therefore our dataset excludes bulls in musth. However, it is possible that some females were pregnant because pregnancy can only be detected about one year into it and no ultrasound was available. Nevertheless, we only had 1 pregnant female and 15 lactating females with calves at heel included in our dataset (we added this information to section 2.2, ll. 185-186 and in the discussion, l. 447). We don’t think these individuals would affect our general results. It would be a very good idea to include data on testosterone when studying reproduction and GC levels, and we actually plan to do so in a future study. Testosterone samples have been collected, but still need to be analysed.
Line 387: at least one?
A: Corrected (L. 434).
Line 661: confounding variables are not in italics
A: Corrected (ll. 277).
Table 2: It would be useful to see the effect size/coefficient for each variable (perhaps as another column) – we can see whether there is a significant difference in e.g. FCM and H/L, but not the direction. Please also ensure that the number of decimal places used in the reporting of statistics is consistent. It would also be useful to indicate the reference category used so that the tables can be interpreted without needing to refer back to the methods (this information also needs to be added there).
A: We understand that presenting effect sizes in table could provide a more holistic view of our results. However, we present the necessary estimates in the results section, and we would prefer to not report them in the table again. As the reviewer says, the reference category would need to be added and several effect sizes for the each other category of each factor would need to be provided in the table, which makes the table, in our view, less clear. However, if the editor decides it would be a better choice to include all these estimates in the table as well we can do so in a revised version.
A couple of other suggestions that could be added to the discussion if this technique is to be used more widely as a measure of stress in individual elephants:
1) care should be taken when assessing blood smears due to different forms of segmented monocytes being categorised as lymphocytes. There are some good reference texts out there, so it will be important to provide criteria (and training) if this technique is to be used more widely, to ensure this is a robust and repeatable measure.
A: We agree with the reviewer. There can be e.g. confusion due to bilobed and sometimes trilobed cells that have been wrongly identified as lymphocytes. Silva & Kuruwita 1993 J Zoo Wildl Med. 24; 434–439 analysed these cells and saw that their granules were peroxidase positive and identical to the granulocyte (neutrophil and monocyte), but did not stain in Leishman-stain blood smears, similar to the non-bilobed monocyte. With the Leishman-stain the cytoplasmic granules of neutrophils are coloured, and so these cells could only be monocytes and not lymphocytes. Similar findings were described by Salakij et al. 2005 Katseart J. (Natural Sci.) 39; 482–493 who used Sudan Black B, α-naphthyl acetate esterase and β-glucuronidase to show that the bilobed and trilobed cells stain in a very similar way to non-bilobed monocytes. We added a sentence to our conclusions that staff training will be necessary if H/L ratios are to be used more widely as a measure of stress (ll. 455-458).
2) What about looking at changes within individuals over time? We know CBCs can be highly individual in this species, so from a clinical perspective, we assess changes against each individual’s reference intervals rather than one produced at a species level – is this something that could be done here too – to look at changes in welfare over time in response to management changes etc.?
A: We are happy about the reviewer’s comment and suggestion. This is an excellent idea, thank you. Our research team has already collected some years’ worth of data on several health parameters and indeed a manuscript on the populations’ range values of these health parameters is currently in press. With more repeated data in the future we hope that we are able to address what the reviewer suggests. It would be very interesting and useful to follow the health parameters (and FGM levels and H/L ratios) of individual elephants over a longer time period, so that we might be able to focus more on the individual’s health and welfare for improved management.
Reviewer 2 Report
There is merit to this paper and you have investigated a useful area of Asian elephants welfare. I suggest making the following changes to the manuscript to improve its clarity and the validity of your results. I would also have emphasized the importance of the ethical value of the welfare assessment of this animal (in the introduction and in the conclusion). I find quite incomprehensible that elelphants are used as working animals.
Specific feedback.
Line 12-15: Delete, not needed
Line 26-28: Delete, not needed
Line 44: Animals worldwide are kept in captivity for diverse reasons.
Line 44-47: Delete: in zoos for entertainment, education and research, as pets, in research facilities for medical, pharmaceutical or product testing, as livestock for dairy, meat and other animal products, or as working animals for draft, transport, rescue or law enforcement.
Line 64: please add an ethical motivation too
Line 85-88 delete: “The release… with the stressor”.
Materials and Methods
The blood sampling was made with the same criterion for all the animals or not (4 hours after feeding for example....)?
You didn’t think to evaluate the acute phase proteins in this species? the acute phase proteins appear as promising parameters to be used in the evaluation of general health status and welfare of animals.
Discussion: Since you have analyzed only two parameters, I think it is worth discussing better the mechanism of action that leads to a variation of H/L ratios, you have mentioned many papers (maybe too many) use them to discuss this issue.
Author Response
There is merit to this paper and you have investigated a useful area of Asian elephants welfare. I suggest making the following changes to the manuscript to improve its clarity and the validity of your results. I would also have emphasized the importance of the ethical value of the welfare assessment of this animal (in the introduction and in the conclusion). I find quite incomprehensible that elelphants are used as working animals.
A: We thank the reviewer for their positive feedback. Regarding the fact that elephants are used as working animals, we, as authors and researchers, feel that we are not in the position to judge and condemn the use of elephants as working animals in their range countries. The human-elephant relationship has a long history and tradition in these countries. Even though certain practises especially in the tourist industry are doubtful and not wholesome for the elephants’ welfare, this is very context-specific and multifaceted issue. However, it is worth noting that due to the semi-captive keeping system elephants employed by the Myanma Timber Enterprise (MTE) show mortality rates, reproductive profiles and social behaviours that resemble those of wild elephants and MTE elephants can forage freely at night and express a large range of their natural behaviours and social interactions in their natural habitat. We address this also in our manuscript (e.g. ll. 156-158, ll. 160-163). We added a sentence about the ethical value of assessing welfare in animals in general in the introduction (ll. 61-62) and in elephants specifically in the discussion (ll. 436-438).
Specific feedback.
Line 12-15: Delete, not needed
A: Deleted as suggested (l. 12).
Line 26-28: Delete, not needed
A: We prefer to keep this sentence as an introduction to our abstract and not to start straight with our aims (ll. 24-26).
Line 44: Animals worldwide are kept in captivity for diverse reasons.
A: We assume this comment refers to the next comment and we kindly refer to our next answer.
Line 44-47: Delete: in zoos for entertainment, education and research, as pets, in research facilities for medical, pharmaceutical or product testing, as livestock for dairy, meat and other animal products, or as working animals for draft, transport, rescue or law enforcement.
A: We would like to keep this information here, since we think it provides a quick overview over the wide range for the reasons why animals are kept in captivity (ll. 42-45).
Line 64: please add an ethical motivation too
A: We added an ethical motivation to the sentence (ll. 61-62).
Line 85-88 delete: “The release… with the stressor”.
A: We prefer to keep this basic information, to make the sentence more intelligible (ll. 85-87).
Materials and Methods
The blood sampling was made with the same criterion for all the animals or not (4 hours after feeding for example....)?
A: Yes, blood sampling followed the same protocol, with samples always taken in the morning after the elephants had been collected from the forest where they feed naturally during the nights. We added this information (ll. 204-205).
You didn’t think to evaluate the acute phase proteins in this species? the acute phase proteins appear as promising parameters to be used in the evaluation of general health status and welfare of animals.
A: So far we have not measured acute-phase proteins and the only measure we have from blood samples are total protein. We thank the reviewer for their very useful comment and we will have a closer look at acute-phase proteins and their link to health and welfare in elephants in the future.
Discussion: Since you have analyzed only two parameters, I think it is worth discussing better the mechanism of action that leads to a variation of H/L ratios, you have mentioned many papers (maybe too many) use them to discuss this issue.
A: We analysed two measures of stress (FGM concentration and H/L ratios) because these are often the main measures of stress in animals (as mentioned in the introduction) and because these were of specific interest in this study. We already discuss some of the potential factors affecting H/L rations “…that GCs in general can have multiple effects on the immune system […] specifically the relative numbers of different types of leucocytes” (ll. 345-347) and that “Various sources of social and physical stress can influence the H/L ratio and the H/L ratio has therefore been suggested to be a valuable measure of stress in vertebrates…” (ll. 347-349). We added some more information on further factors that can influence the H/L ratio (ll. 443-446) However, the underlying molecular mechanisms leading to differences in H/L ratios between individuals are well summarized in the reviews and studies we cite and we think it is better if interested readers should refer to those publications, because this is not the specific focus here and we have not studied such mechanisms in our article.
Reviewer 3 Report
The paper submitted by Seltman et al. (669558) reports on the use of heterohil / lymphocytes ratio as a measure of stress in Asian timber elephants.
There are a few things that require attention:
Please check your terminology, I don’t think it is correct to refer to specific white blood cells in elephants as heterophils. It would also be better to refer to faecal glucocorticoid metabolite concentrations, as it is not stated which specific steroid metaboiltes (cortisol or corticosterone) were measured.
Line 85: correct faeces samples to faecal samples
Line 97: please check the terminology
Methods:
Specify which assay was used to measure faecal glucocorticoid metabolites and how it was validated.
Was a parallelism test performed to validate the accuracy of the chosen assay?
Line 186: are these values referring to wet or dry weight?
Line 186: The absolute values FCM concentrations cannot be compared between studies, especially not when it is not specified which steroid metabolites have been used for either of them
Results:
3.6: Please specify how many data points of body weight you have per individual and if the given example of lost body weight results from 1 individual or is cross sectional.
Instead of Figure 1b a standardised residual plot should be shown to assess the fit of the model.
Author Response
The paper submitted by Seltman et al. (669558) reports on the use of heterohil / lymphocytes ratio as a measure of stress in Asian timber elephants.
There are a few things that require attention:
Please check your terminology, I don’t think it is correct to refer to specific white blood cells in elephants as heterophils. It would also be better to refer to faecal glucocorticoid metabolite concentrations, as it is not stated which specific steroid metaboiltes (cortisol or corticosterone) were measured.
A: The reviewer is correct when they refer to mammals usually having neutrophils instead of heterophils. However elephants, like all members of the superorder Afrotheria, have heterophils instead of neutrophils. We added this information in the introduction (ll. 114-119) (please see also comment 1 from reviewer 1). Following the suggestion of the reviewer we changed all cases of faecal cortisol metabolites to faecal glucocorticoid metabolites.
Line 85: correct faeces samples to faecal samples
A: Corrected (l. 85).
Line 97: please check the terminology
A: We changed the sentence to “Heterophils (or neutrophils, depending on the species) and lymphocytes constitute…” (ll. 98-99).
Methods:
Specify which assay was used to measure faecal glucocorticoid metabolites and how it was validated.
A: The enzyme immune assay used in this study was developed and validated for use for Asian elephant faecal samples by Watson and colleagues (Watson et al. 2013 General and Comparative Endocrinology 186; 16–24), please see line 196 for the corresponding reference in our methods. Our collaborators at the Veterinary Diagnostic Laboratory at Chiang Mai University in Thailand have long experience using this specific assay on faecal samples from Asian elephants from our study population and other ones.
Was a parallelism test performed to validate the accuracy of the chosen assay?
A: Yes, the assay was biochemically and biologically validated, also using parallel displacement curves. Please see e.g. in the abstract of Watson et al. 2013 General and Comparative Endocrinology 186; 16–24 “Following optimization, this EIA was then validated biochemically for 38 species, through parallel displacement curves and interference assessment tests of faecal and urine samples. Additionally, biological validation was per- formed opportunistically in a subset of species, with use of this EIA demonstrating significant elevations in faecal glucocorticoid metabolites following potentially challenging events”
Line 186: are these values referring to wet or dry weight?
A: These values are referring to dry weight, please see also line 192 (“…until drying in a hot air oven at 50°C. Dried samples were shipped for further analysis…”).
Line 186: The absolute values FCM concentrations cannot be compared between studies, especially not when it is not specified which steroid metabolites have been used for either of them
A: We agree with the reviewer that comparisons between FGM concentrations between different studies is difficult and sometimes not advisable. However, we think it is still useful to report these values here, to get an estimate of variation of FGM concentrations within our study population. Again, when discussing FGM concentrations between different studies, this should be done very carefully paying attention to different methods, environmental conditions and life-history stages of the investigated animals. We added this to the corresponding section (ll. 199-200).
Results:
3.6: Please specify how many data points of body weight you have per individual and if the given example of lost body weight results from 1 individual or is cross sectional.
A: For the body weight dataset we have on average two measurements (with matching body weight, FGM and H/L ratio) per individual (see lines 219-220), where the number of measures ranged from 1 to 8 per individual (see lines 219-220). The given example in the result section has been calculated based on our results to demonstrate the average impact of cortisol on body weight of an adult elephant.
Instead of Figure 1b a standardised residual plot should be shown to assess the fit of the model.
A: We would prefer to keep figure 1b as it is and not replace it with a standardised residual plot. We checked our data distribution and did examine a residual plot to make sure our model fits well. However, these are for basic model diagnostics (See also lines 227—228) and we don’t think an additional plot in the manuscript will give more insights, but we mention these results in the methods section (see line 241-242). We also included a residual QQ plot indicating model fit below (see figure S1, please see attachment). If the reviewer and editor wish so, this plot could be included to the manuscript as a supplementary file.
Figure S1. A residual QQ plot indicating model fit for model 1 (see Table 2 in the manuscript).
